# Polyunsaturated Aldehydes Profile in the Diatom *Cyclotella cryptica* Is Sensitive to Changes in Its Phycosphere Bacterial Assemblages

**DOI:** 10.3390/md21110571

**Published:** 2023-10-30

**Authors:** María Hernanz-Torrijos, María J. Ortega, Bárbara Úbeda, Ana Bartual

**Affiliations:** 1Instituto Universitario de Investigaciones Marinas (INMAR), Campus de Excelencia Internacional del Mar (CEI-MAR), Universidad de Cádiz, Puerto Real, 11510 Cádiz, Spain; maria.hernanz@uca.es (M.H.-T.); barbara.ubeda@uca.es (B.Ú.); 2Departamento de Biología, Facultad de Ciencias del Mar y Ambientales, Universidad de Cádiz, Puerto Real, 11510 Cádiz, Spain; 3Departamento de Química Orgánica, Facultad de Ciencias del Mar y Ambientales, Universidad de Cádiz, Puerto Real, 11510 Cádiz, Spain; mariajesus.ortega@uca.es

**Keywords:** diatom–bacteria interaction, infochemical, PUA, PUFA

## Abstract

Diatoms are responsible for the fixation of ca. 20% of the global CO2 and live associated with bacteria that utilize the organic substances produced by them. Current research trends in marine microbial ecology show which diatom and bacteria interact mediated through the production and exchange of infochemicals. Polyunsaturated aldehydes (PUA) are organic molecules released by diatoms that are considered to have infochemical properties. In this work, we investigated the possible role of PUA as a mediator in diatom–bacteria interactions. To this end, we compare the PUA profile of a newly isolated oceanic PUA producer diatom, *Cyclotella cryptica,* co-cultured with and without associated bacteria at two phosphate availability conditions. We found that the PUA profile of *C. cryptica* cultured axenically was different than its profile when it was co-cultured with autochthonous (naturally associated) and non-autochthonous bacteria (unnaturally inoculated). We also observed that bacterial presence significantly enhanced diatom growth and that *C. cryptica* modulated the percentage of released PUA in response to the presence of bacteria, also depending on the consortium type. Based on our results, we propose that this diatom could use released PUA as a specific organic matter sign to attract beneficial bacteria for constructing its own phycosphere, for more beneficial growth.

## 1. Introduction

Diatoms are one of the most important and diverse groups of phytoplankton widely distributed in oceanic waters. They form the basis of the marine food web of most aquatic ecosystems, being able to fix nearly 20% of global carbon on Earth [1]. Each diatom cell, as a consequence of its biological activity, generates a region in its immediate environment, the phycosphere, characterized by being enriched in dissolved organic compounds released or exuded by it [2]. This dissolved organic matter (DOM) can influence the growth and metabolism of other associated organisms in its direct environment [3] but in particular, is host to diverse microbial assemblages that thrive in close proximity to the diatom, either attached or free-living [4]. Interactions between diatoms and bacteria are some of the most important relationships in aquatic environments [3,4,5]. These interactions are diverse and complex and can result in mutualistic, commensal, competitive, and antagonistic interactions that can lead to the demise or success of interacting species [5,6,7]. Although the precise characteristics of these interactions may exhibit variability, chemical compounds emerge as the main drivers of these dynamics [8,9,10]. 

Polyunsaturated aldehydes (PUA) are organic molecules that make up the phycosphere of various diatoms [11,12]. PUA are long-chain volatile oxylipins, derived from lypoxidation of intracellular polyunsaturated fatty acids (PUFA) [13]. Some authors have demonstrated that, in nature, at the end of a diatom bloom period, PUA are released concomitantly to nutrient limitation [14,15] and nowadays, several authors defend their function as infochemicals regulating diatoms bloom mediated by the bottom-up control of herbivore populations [11,16,17,18,19]. In the open ocean, it has been described a meaningful macroecological relationship with resource availability (unbalanced N:P ratio) shows a higher PUA production capacity in the phytoplankton of the poorest waters and among the small species typically populating these environments [20,21]. Experimentally unbalanced N:P ratios of low P or low N also triggered higher PUA production in diatoms as in [22]; they experimentally demonstrated that low silicon, low P, and low N enhanced PUA production in a coastal diatom. The possibility that PUA may play a role as a mediator of diatom–bacteria interactions in the phycosphere has been less studied and published results are somewhat more equivocal [22,23]. Some authors showed differential effects of PUA on the growth and metabolism of natural free-living bacterial communities [22,23], and Edwards et al. suggested that PUA altered the community structure of particle-associated bacteria suggesting a role in bacterial community succession [24]. Very recently, Eastabrook et al. have suggested that PUA is not the main driver of diatom–bacteria interactions in laboratory cultures [25]. Recent publications also consider the significant role of attached bacteria in phosphorus recycling in nature [26,27]. 

In this work, we want to further analyze PUA as a mediator in diatom–bacteria interactions, by studying the effect that the presence and conformation of microbial consortia might have on PUA production by the diatom. To explore this possible diatom–bacteria interaction, we shifted the microbial consortium present in cultures of *C. cryptica*, a newly isolated PUA producer coastal diatom. We studied the effect of the presence and absence of microbes by comparing the type and amount of PUA produced by the diatom host *C. cryptica* co-cultured with different microbial assemblages and compared with axenic conditions. This study was performed under experimental conditions of N:P imbalance that favored PUA-enriched phycospheres as has been demonstrated in the literature.

## 2. Results

### 2.1. Co-Cultures Assemblages Description

Representative cytograms from the experiments carried out are shown in Figure 1. *C. cryptica* cells were easily distinguished by chlorophyll autofluorescence (Figure 1A) and side scatter (SSC); axenic cultures of *C. cryptica* were checked to remain free of bacteria until the end of the experiment by SYBR Green staining (Figure 1B). Biotic particles (e.g., heterotrophic bacteria) can be distinguished from nonbiotic particles (cellular residues and detritus) by staining with SYBR Green. Positively stained cells showed positive fluorescence (unit ≥ 10^4^) (Figure 1C,D) but did not appear in axenic cultures (Figure 1B).

### 2.2. Growth Rates

*C. cryptica* showed the lowest growth rates under axenic conditions (A-cultures) (Table 1). However, growth rates of *C. cryptica* increased significantly (*p*-value < 0.05) in N-cultures (*C. cryptica* co-cultured with autochthonous heterotrophic bacteria) and I-cultures (*C. cryptica* co-cultured with introduced heterotrophic bacteria). The results showed that the presence of I-cultures positively influenced their growth, reaching the highest growth rates of all experiments (*p* < 0.05) (Table 1). In A-cultures and I-cultures, growth rates were significantly lower under low P conditions compared with F2 culture conditions (one-way ANOVA; *p*-value < 0.05). No significant differences were observed at the two nutrient conditions for N-cultures (one-way ANOVA; *p*-value > 0.05). Regarding growth rates of heterotrophic bacteria, in N-cultures, heterotrophic bacteria showed a significant reduction of growth rates than in I-cultures in both P conditions assayed (one-way ANOVA; *p*-value < 0.001) (Table 1).

### 2.3. PUA

#### 2.3.1. Total pPUA Concentration 

pPUA (particulate PUA) were detected in all sampled cultures, with an average total pPUA concentration ranging from 0.02 to 3.46 fmol cell^−1^ (Table 2, Appendix A). The highest concentrations of total pPUA were recorded in N-cultures under both phosphate availability treatments (one-way ANOVA; *p*-value < 0.001, Table 2). By nutrient availability, pPUA was significantly higher in low P cultures compared with F2 cultures when *C. cryptica* grew axenically and in N-cultures (one-way ANOVA; *p*-value < 0.001). Contrarily, pPUA was significantly lower in low P cultures compared with F2 cultures in I-cultures (Table 2, Appendix A). A two-way ANOVA was performed to evaluate the effect of P availability and bacterial presence in pPUA, and the total pPUA concentration was influenced by bacterial presence, phosphate availability, and their interaction (two-way ANOVA; *p*-value < 0.01, *n* = 5) (Appendix A).

#### 2.3.2. Quantitative and Qualitative Analysis of pPUA Types

Five different types of pPUA were identified in the analyzed samples: particulate *2E,4E/Z*-heptadienal (pHD), particulate *2E,4E/Z*-octadienal (pOD), particulate *2E,4E/Z*-decadienal (pDD), particulate *2E,4E/Z,7*-octatrienal (pOT) and particulate *2E,4E/Z,7Z*-decatrienal (pDT). pHD was generally the most abundant aldehyde in all treatments, and pOT was the least abundant (Table 3). When *C. cryptica* grew axenically pOT and pDT were not detected at F2 conditions, while in the other cultures, pDT was detected in the range of 0.01 to 0.51 fmol cell^−1^ (Table 3). Averages concentrations of different pPUA types are shown graphically in Appendix A.

The presence of both autochthonous (N-cultures) and introduced bacteria (I-cultures) affected the diversity of the types of pPUA produced by *C. cryptica*. When bacteria were present, significant differences were detected in the averages of pHD, pOD, pDD concentrations when compared with A-cultures, at both phosphate availability conditions assayed (one-way ANOVA; *p*-value < 0.05) (Table 3). At a low P condition, statistically significant differences were observed for averaged pDT between axenic cultures and N-cultures (one-way ANOVA; *p*-value < 0.05) (Table 3), but no significant differences were observed for averaged pOT (one-way ANOVA; *p*-value > 0.05) (Table 3). 

By relative percentage of abundance, pHD was significantly lower in I-cultures at both nutrient conditions (one-way ANOVA; *p*-value > 0.05) (Figure 2). The relative abundance of pOD varied significantly in axenic cultures between P conditions, ranging from 31% in the F2 condition to 20% in low P conditions (one-way ANOVA; *p*-value < 0.05 (Figure 2). pDD, pOT, and pDT relative abundances differed significantly among cultures and P conditions. In fact, the qualitative distribution of pPUA was influenced by bacterial presence, phosphate availability, and their interaction (two-way ANOVA; *p*-value < 0.05, *n* = 5).

#### 2.3.3. Total dPUA Concentration

dPUA (dissolved PUA) was detected in all sampled cultures. Under the F2 nutrient condition, total dPUA ranged from a minimal concentration of 2.03 nM in I-cultures to a maximum of 9.72 nM in the N-cultures (Table 4). Under low P conditions, total dPUA concentration increased significantly from 4.81 ± 0.32 nM in N-cultures to 8.39 ± 2.16 nM in axenic cultures (one-way ANOVA; *p*-value < 0.01) but decreased significantly from 9.72 ± 2.13 nM to 4.81 ± 0.32 nM in N-cultures (one-way ANOVA; *p*-value < 0.01) (Table 4, Appendix A). In I-cultures, no statistically significant differences were observed in average total dPUA between F2 and low P conditions (one-way ANOVA; *p*-value = 0.109) (Appendix A). Statistically significant differences were observed for averaged total dPUA between axenic and non-axenic cultures, at the two nutrient availability conditions assayed (one-way ANOVA; *p* < 0.05) (Table 4). A two-way ANOVA was performed to evaluate the effect of P availability and bacterial presence in dPUA, showing that total dPUA concentration was affected by the bacterial presence and the interaction of bacterial presence and phosphate availability (two-way ANOVA; *p*-value < 0.001; *n* = 5; Appendix A.

#### 2.3.4. Quantitative and Qualitative Analysis of dPUA Types

Five different dPUA were identified and quantified, dissolved *2E,4E/Z*-heptadienal (dHD), dissolved *2E,4E/Z*-octadienal (dOD), dissolved *2E,4E/Z*-decadienal (dDD), dissolved *2E,4E/Z,7*-octatrienal (dOT) and dissolved *2E,4E/Z,7Z*-decatrienal (dDT) (Table 5). We found that dHD was generally the most abundant dPUA in most conditions assayed with a range between 1.65 and 4.01 nM (Table 5).

The presence of both, autochthonous and introduced bacteria, significantly affected the diversity of dPUA, compared with the proportions detected in the axenic *C. cryptica* cultures, in both phosphate availability conditions (one-way ANOVA; *p*-value < 0.05) (Table 5, Figure 3). When *C. cryptica* grew axenically (A-cultures) and in N-cultures, we observed significant variations in dPUA types when the relative abundances under F2 and low P conditions (one-way ANOVA; *p*-value < 0.05). However, the diversity of dPUA detected was very low in I-cultures and only dHD and dDT were detected (Figure 3). In I-cultures, nutrient conditions did not affect such trends, and no significant differences could be established in phosphate availability (one-way ANOVA; *p* > 0.05). Qualitative changes in dPUA were influenced by the bacterial presence, phosphate availability, and their interaction (two-way ANOVA; *p*-value < 0.05, *n* = 5). Detailed average dPUA-type concentrations detected in the different cultures of *C. cryptica* are shown graphically in Appendix A.

A matrix correlogram was performed for each nutrient availability condition (Figure 4A,B). Under the F2 condition (Figure 4A), the total abundance of bacteria, dead bacteria, and alive bacteria were positive and significantly correlated with total pPUA, total dPUA, and all the PUA types (r > 0.5; *p*-value < 0.05), with the exception of dOD and dDD (*p*-value > 0.05). Under low P conditions (Figure 4B), total heterotrophic bacteria were negative and significantly correlated with total pPUA and all the pPUA types (r < −0.5; *p*-value < 0.05). Regarding dPUA, at low P conditions, dead bacteria were negative and significantly correlated with dPUA and dOD (r = −0.81; r = −0.75, respectively; *p*-value < 0.05) and alive heterotrophic bacteria were negative and significantly correlated with dDD and with dDT (r = −0.80 and r = −0.77, respectively; *p*-value < 0.05) (Figure 4B). Without nutrient distinction and considering all data sets (Figure 4C), we can discern strong positive correlations of total bacteria and alive bacteria with total pPUA and every type (r > 0.8; *p*-value < 0.05). Alive bacteria were positively correlated with dHD and dOT and dead bacteria were negatively correlated with dOD and dDD (Figure 4C).

#### 2.3.5. Percentage of Released PUA (%dPUA)

The summary of total pPUA and total dPUA can be assumed as the total pool of PUA in the culture at every assayed condition and time. Considering that only diatoms are the source of dPUA, we can estimate the relative proportion of “released” PUA in the medium with respect to the total pool of PUA as follows:$$\frac{\left[\mathrm{dPUA}\right]}{\sum \left[\mathrm{pPUA}\right]+[\mathrm{dPUA}]}\;\times \;100\;$$

At F2 conditions, the highest % of released dPUA (30%) was recorded at axenic conditions, however, under low P conditions the highest %dPUA (42%) was found in I-cultures. The lower %dPUA was reported in N-cultures under both phosphate availability treatments (Figure 5). Significant differences in %dPUA were detected between axenic and non-axenic cultures at both nutrient conditions (one-way ANOVA; *p*-value < 0.001) and also between non-axenic cultures at the two P conditions (one-way ANOVA; *p*-value < 0.05). Comparing between F2 and low P conditions, we found significant differences in the three experiments. In A-cultures and in N-cultures, %dPUA decreased significantly in low P condition (one-way ANOVA; *p*-value = 0.001) (Figure 5). On the contrary, in I-cultures, %dPUA increased under low P condition (one-way ANOVA; *p*-value = 0.001) (Figure 5). %dPUA was significantly affected by the bacterial presence, phosphate availability, and their interaction (two-way ANOVA; *p*-value < 0.05, *n* = 5). PUA types, instead of total PUA, are shown in Appendix A.

### 2.4. Fatty Acid Composition

Since the highest concentrations of total pPUA were recorded when *C. cryptica* was cultured with autochthonous bacteria and it was nutrient dependent (Table 3, Appendix A), we analyzed the FAME (fatty acid methyl ester) profiles of the stock cultures of *C. cryptica* under F2 and low P conditions and in two different growth phases. Analysis by GC-MS displayed, in F2 conditions, that C16 and C20 fatty acids were the most abundant, both in exponential and late exponential growth phase with medium values of 65.75 and 17.91 fmol cell^−1^ for exponential phase and 93.63 and 34.70 fmol cell^−1^ for late exponential phase (Appendix A). These results showed for these fatty acids a general increase in concentration at the late exponential phase. For low P conditions, the analysis also revealed high concentrations of C16 and C20 fatty acids (165.42 and 19.89 fmol cell^−1^, respectively). It was also observed a remarkable amount of C18 fatty acids with 21.99 fmol cell^−1^. However, unlike the F2 conditions, in the late exponential growth phase, these values decreased to 54.40, 1.41, and 3.64 fmol cell^−1^ for C16, C18, and C20 fatty acids, respectively. These findings demonstrate a decrease in fatty acids under low P conditions in the late exponential growth phase.

In the FAME analysis the detected PUFA included the main polyunsaturated diatom markers as C16:2, C16:3, and C16:4 along with other C18 acids (linoleic, α-linoleic, γ-linoleic acids, and C18:4), the C20 fatty acids 20:3, arachidonic acid (ARA), and eicosapentaenoic acid (EPA), and the C22 docosahexaenoic acid (DHA) (Figure 6). Linolelaidic acid, eicosatrienoic acid, and C16:2 were present only in low P conditions. In such conditions, C16:4 and ARA were present at trace levels. The major PUFA present in both phosphate availability conditions was EPA. Specifically, in F2 conditions, EPA represents a range of 14.90–21.92% from the exponential to late exponential growth phase, while in low P conditions, the medium value of EPA in both growth phases was 8.97% (Table 6). Other relevant PUFAs in F2 conditions were C16:3 (17.09–14.30%) and docosahexaenoic acid (2.93–4.31%), while in low P conditions, other remarkable PUFAs were C16:2 (3.22–2.12%), C16:3 (3.29–1.73%), and DHA (2.6–2.24%) (Figure 6).

In summary, the number of fatty acids increased in F2 conditions as cultures progressed to the late exponential growth phase, ranging from 116.71 to 154.25 fmol cell^−1^. However, in low P conditions, although the number of fatty acids in the exponential phase (216.13 fmol cell^−1^) exceeded that observed in F2 conditions (116.71 fmol cell^−1^), it significantly decreased when cultures reached the late exponential phase, reaching 41.60 fmol cell^−1^. Notably, in low P, key polyunsaturated diatom acids markers such as C16:2 and C16:3 declined in the late exponential growth phase, while EPA and DHA, representative fatty acids among the ω-3 pool in *C. crytica* maintained their percentages in both growth phases. Thus, low phosphorous conditions [28,29] led to the accumulation of lipids in the exponential growth phase mainly in C16:0, C18:0, and C16:1 at the expense of polyunsaturated fatty acids.

## 3. Discussion

As the first significant result in this study, we have observed that bacteria enhanced the growth rate of *C. cryptica* (Table 1). Axenic cultures of *C. cryptica* (A-cultures) showed significantly lower exponential growth rates (0.35–0.42 day^−1^) compared with N-cultures (0.53–0.62 day^−1^) and I-cultures (0.73–0.82 day^−1^) at both nutrients conditions (Table 1). Furthermore, a significant increase in growth rates of *C. cryptica* was observed when non-native bacteria were inoculated (I-cultures) reaching the highest growth rates (0.82 ± 0.21 day^−1^). These results evidenced that re-inoculation of the microbiota community associated with a different diatom (particularly, microbiota associated with *P. tricornutum*) had an ultimate benefit on *C. cryptica* growth. 

The observed beneficial effects of bacteria on diatom growth are not new. Since Provasoli first suggested that bacteria could enhance algal growth [30], many studies have corroborated this hypothesis and it is now well known that diatoms develop specific interactions with certain bacteria that conform microbial consortia, both in culture and in nature (reviewed in [31]). Sometimes these interactions are almost forced, and bacterial activity satisfies a particular diatom requirement (e.g., vitamins or iron) [32,33]. Further indirect support for this view comes from the frequent observation that prolonged culturing of diatoms in the absence of bacteria negatively influences their physiology and growth [34,35]. What is interesting about our results is that, along with this beneficial effect of bacteria on *C. cryptica* growth rates, we have also observed a significant effect of bacteria on diatom pPUA (Table 2 and Table 3; Appendix A) and dPUA profiles (Table 4 and Table 5; Appendix A), denoting a striking role of PUA in diatom–bacteria interaction. 

In N-cultures, the higher pPUA concentration observed at low P cultures (higher N:P imbalance) (Table 2) is consistent with the literature [14,21,36]. We can establish the pPUA profile obtained in N-cultures, with HD (>50%), OD (>15%), and DT (>9%) (Figure 2) as a representative pPUA profile of *C. cryptica* adapted to our experimental environmental conditions. These PUA have been well documented as the dominant bioactive PUA released by diatoms in the past [11,12,13,36,37,38,39,40]. Also, this PUA profile is consistent with the cellular PUFA content observed in stock cultures of this study (Figure 6) as main PUA precursors: eicosapentaenoic acid (EPA, C 20:5) as HD and DT precursor, C16:3 and C16:4 as OD and OT precursors and C 20:4 as DD precursor [13,41,42]. Cells from N-cultures, showed the highest pPUA concentrations, compared with I-cultures and A-cultures (Table 2), especially at low P. However, the estimated percentage of PUA released (%dPUA in Figure 5) for N-cultures was very low (<3%) (Figure 5) at both nutrient conditions when compared with A-cultures (up to 30% at F2) and I-cultures (up to 45% at low P). Furthermore, for N-cultures, dOT was also a representative PUA (≥20%) at both nutrient conditions (Figure 3), which was not proportionally abundant in the pPUA of the cells (Figure 2). This can be interpreted in two ways. Firstly, diatom cells at N-cultures have all the metabolic requirements to synthesize PUA (N-cultures in Table 2; PUFA in Figure 6) but they do not release all potential PUA to the media, resulting in a low % dPUA (Figure 5). Secondly, cells at N-cultures released a higher concentration of dPUA to the media but the released PUA is used as a source of carbon by their associated autochthonous bacteria, and then, we underestimated dPUA concentration and diversity in the media. We found that dPUA was positive and significantly correlated with bacterial abundance at F2 conditions (Figure 4A) but not at low P conditions (Figure 4B). In addition, we found a differential effect of bacteria on the type of dPUA, as indicative that bacteria could use dPUA as a carbon source in a selective manner. It seems that introduced bacteria (I-cultures) use all dOD, dDD, and dOT (Figure 3) to significantly increase their growth (Table 1). As a result, this would alter the proportion of PUA types in the dissolved fraction compared to pPUA as observed by comparing Figure 2 and Figure 3, and furthermore, this is consistent with the growth observed for bacteria (Table 1). PUA would help to maintain a stable diatom–bacteria association by conforming the community structure, in fact, some PUAs are positively correlated with total and alive bacteria concentration and others with dead bacteria (Figure 4C). This is consistent with previous works that found a differential effect of PUA on marine bacteria [22,23]. Several authors have shown that diatoms conserve associated bacterial communities across time [43], therefore, associations are not randomly assembled but follow a dynamic that can be reproduced [44,45].

Starting from the premise of considering N-cultures as representative of this diatom in this particular culture conditions, if we compare with A-cultures, the relative % of PUA released increased greatly in axenic conditions (%dPUA in Figure 5), especially at F2, where precisely lower concentrations of pPUA were quantified in the cells (Table 2), and growth rates were the lowest (Table 1). That is, axenic cells with lower pPUA (A-cultures in Table 2) released the highest percentage of dPUA at both nutrient conditions (Figure 5). Ultimately, this resulted in the lowest observed diatom growth rates (A-cultures in Table 1). In these cultures, any effect of bacterial presence on the quantified dPUA can be ruled out since cultures were axenic. Interestingly, when non-native bacteria (from a non-PUA producer diatom) were added to the axenic cultures (I-cultures), the dPUA released reached the highest value under low P (45% in Figure 5), resulting in this case, in a better diatom growth rate (Table 1) under both nutrient conditions. Even taking into account a possible bacterial use of dPUA as a source of carbon, the %dPUA was the highest observed. It is important to note that *C. cryptica* cells of I-cultures come from A-cultures re-inoculated with non-native bacteria, and their pPUA should be similar, as is corroborated in Table 2. 

PUA released at low P ultimately produced a benefit in diatom growth, and, in the case of introduced bacteria (I-cultures) also bacteria grew optimally (Table 1). Thus, we can infer an apparent mutual benefit in the diatom–bacteria growth in which PUA participates in some way. We did not analyze the diversity of associated microbiota for each experiment. Released PUA could lead to changes in the diversity of the bacterial community for N-cultures and I-cultures, as it has been reported by several authors for coastal bacteria [23], or not, as has been documented by Eastabrook et al. [25], for laboratory diatom cultures, but this does not detract from the fact that we found a mutual benefit for growth. We hypothesize that, albeit at nM levels, the PUA released (type and quantity) could contribute to defining a specific organic matter signature of this diatom at each assayed condition. These molecules together with other organic substances would help to conform to a particular phycosphere during bacterial colonization of the cell vicinity. This could help to design a specific bacterial niche that would confer an advantage for diatom growth, as we have observed in the presented results. The associated bacterial community would be relatively stable over time mainly in experimental cultures maintained by successive reculturing. In the case of A-cultures, where diatom cells have the lower pPUA, they released the most PUA (highest %dPUA in Figure 5). This can be understood as a failed attempt to attract bacteria. In A-cultures, where there are no bacteria, PUA release is an unsuccessful strategy of diatom cells to attract non-existent bacteria, with a significant metabolic expenditure that eventually takes its toll, and *C. cryptica* growth slows. 

In this work, we have been focused on the analysis of diatom PUA, but also bacteria can release molecules that interact with diatoms and might influence the excretion of algal metabolites (including PUA). It has been demonstrated that hormones from bacteria can enhance the cell division of algal cells, its photosynthetic machinery, and potentially its carbon output to the bacteria [31,46]. Other bacteria manipulate algal growth by producing proteins that lyse algal cells or unknown factors that arrest algal cell division [47,48,49,50]. Then, we cannot conclude if any of the quantified PUA is upregulated in response to signals from co-cultured bacteria or if diatoms can take up metabolites released by bacteria. In fact, an uptake of any compound released from the bacteria could influence specific PUFA/PUA biosynthetic pathways of the algae [3,32]. This would need a new and particular experimental design focused on it. 

Our results demonstrated that *C. cryptica* modulated its PUA profile in response to the presence or absence of bacteria in the surrounding media, conferring a growth advantage. It is an emerging concept that marine microbial communities are part of tightly connected networks, providing evidence that these interactions are mediated through the production and exchange of infochemicals [31]. These inter-kingdom interactions are complex, involving the exchange of cofactors, micronutrients, macronutrients, proteins, and signaling molecules, and PUA could be an active part of the variety of molecules involved.

## 4. Materials and Methods

### 4.1. Biological Material

Two diatom species were used in this study: the PUA producer *Cyclotella cryptica* and the non-PUA producer *Phaeodactylum tricornutum* (strain CCAP 1052/1A). *C. cryptica* was freshly isolated from Atlantic coastal area at the Bay of Cádiz (Southwestern Spain). The species was identified by PCR using the protocol described by Zimmermann et al., at the University Institute of Marine Research (INMAR, Cádiz, Spain) [51]. Before starting the experiments, we checked that PUA was not detected in monocultures of *P. tricornutum* and its associated heterotrophic bacteria (introduced bacteria) by using monocultures of this diatom, following the protocols explained below (Section 4.5). Therefore, it can be ensured that the PUA detected in all the experiments was exclusively produced by *C. cryptica.*

*C. cryptica* and *P. tricornutum* stock cultures were maintained by successive inoculations at constant 20 °C and of 14:10 L:D cycle in sterile natural seawater enriched with f/2 medium [52] and silicate. Additionally, an inoculum of the isolated strain of *C. cryptica* was purified by Spanish Algae Bank (https://marinebiotechnology.org/en/), by fluorescence-activated cell sorting (FACs) using a Sony SH800 flow cytometer (Tokyo, Japan). combined with serial dilutions. First, diatom cells were collected from a non-axenic culture of *C. cryptica* using cell sorting option of the cytometer. Separated cells were inoculated one by one into a 96 multi-plate with sterile ASP12 medium. Once colonies were detected, cells were collected and transferred to Erlenmeyer flasks following [53] protocol, at 20 °C, 50 μmol quanta m^−2^ s^−1^ in a 14:10 L:D cycle. Diatom populations with lower bacterial cell density were washed by successive centrifugation and dilutions until axenic conditions were observed. A stock culture of this axenic strain was also maintained, and the absence of bacteria was periodically verified by examining SYBR Green I nucleic acid gel-stained cultures by imaging flow cytometry (IFC) using a Luminex ImageStream^®^X Mk II [54] (Seattle, WA, USA). 

### 4.2. Experimental Design

Axenic and non-axenic cultures of *C. cryptica* were grown in 250 mL sterilized Erlenmeyer flasks under artificial light at 54 ± 6 μmolquanta m^−2^ s^−1^ of irradiance and 14:10 light:dark (L:D) cycle. They were maintained at 20 °C and orbital agitation at 90 r.p.m. under two nutrient conditions: The first one was obtained by culturing the axenic and non-axenic strains of *C. cryptica* in sterilized 0.22 μm filtered natural seawater enriched with f/2 nutrient stock and silicate [52], with a final phosphate concentration of 32.42 ± 0.57 µM (hereafter, F2). For the second one, 0.22 μm filtered natural seawater enriched with a lower final concentration of phosphate of 5.55 ± 0.16 µM (hereafter low P). Nitrate concentration was 963.45 ± 0.53 µM and silicate was 106 ± 1.01 µM for both treatments. These nutrient concentrations were quantified spectrophotometrically with a Skalar autoanalyzer following standard procedures of Strickland and Parsons [55], from samples filtered through pre-combusted Whatman GF/F filters (200 °C; 4 h). Five replicates per treatment were followed. 

To assess the potential qualitative and quantitative impact of microbial presence on PUA production, three sets of experiments were carried out: one using axenic cultures of *C. cryptica* (hereafter, A-culture, axenic), second, non-axenic cultures of *C. cryptica* co-cultured with the heterotrophic bacteria naturally associated (hereafter, N-culture, natural heterotrophic bacteria), and third, *C. cryptica* co-cultured with bacterial communities associated to the non-PUA producer diatom *P. tricornutum* (hereafter, I-culture, introduced heterotrophic bacteria). Introduced heterotrophic bacteria were obtained from a dense stock culture of *P. tricornutum*. This culture was filtered through sterilized polycarbonate filters (Whatman^®^-Nucleopore^TM^ Track-Etch Membrane, Maidstone, UK) with 2.0 μm of pore size to remove algae while preserving the bacterial communities associated with *P. tricornutum*. Each experimental flask (*n* = 5) containing axenic *C. cryptica* cells, was inoculated with an introduced bacterial inoculum of 6 × 10^3^ cells mL^−1^. 

Daily samples were collected from each flask for quantification of diatom cell density under sterile UV ambient. Cultures were monitored from the inoculation day until late exponential growth phase. For the axenic experiments, samples for cell density quantification were only collected on the inoculation day and ending day to avoid any contamination. Once the cultures reached the late exponential growth phase, samples for particulate and dissolved PUA (hereafter, pPUA and dPUA) were collected. Also, samples for analyzing the PUFA profiles of the stock cultures of *C. cryptica* under F2 and low P conditions were collected. For clarification, a detailed flow chart of this experimental design is provided in Appendix A. 

### 4.3. Cell Density Quantification

Diatom and bacterial cell densities were quantified by image flow cytometry (IFC) using a Luminex ImageStream^®^X Mk II (hereafter, ISX, Seattle, WA, USA). IFC enables multimode imaging of cells simultaneously in bright field, dark field (analogous to flow cytometry measured side scatter laser light, SSC), and a broad range of fluorescence wavelengths using a time delay integration charge-coupled device (CCD) camera that integrates images passing through the field, generating a high-resolution summed image associated to each particle [56]. The ISX used was equipped with 60×/40×/20× magnifications and four excitation lasers (405 nm, 488 nm, 642 nm, and 785 nm). Samples were analyzed at 60× magnification and low flow rate, after excitation with 488 and 785 nm lasers using the INSPIRE software (2023.2.173.0) (Amnis Corp., Seattle, WA, USA). For each sample, a range between 10,000–20,000 particles was collected. Side scatter images (SSC) were obtained using 785 nm excitation and 745–800 nm emission and chlorophyll autofluorescence (642–745 nm emission) was detected with 488 nm excitation laser. In addition, a CCD camera collected images in the bright field channel (BF) associated with each particle suspended in the analyzed sample. Post-acquisition spectral compensation and data analysis were performed using the IDEAS 6.2 image analysis software package (Amnis Corp., Seattle, WA, USA). Figure 7 shows representative dot plots obtained with associated images at 60x magnification to each dot. By combining dot plots of different channels of the ISX, suspended particles in the sample can be discerned, and images of each particle can be analyzed with IDEAS 6.2. image analysis software. The chlorophyll *a* in autotrophic alive cells allows these cells to be discerned by autofluorescence of this pigment after blue light excitation. As shown in Figure 7, alive *C. cryptica* cells were easily localized in the obtained cytograms by chlorophyll autofluorescence and SSC (Figure 7A), and the images (60× magnification) associated with each cell were collected (Figure 8). For quantification of non-autofluorescent cells as heterotrophic bacteria, flow cytometric counting was possible through the staining of cell nucleic acids with fluorescent dyes prior to analysis. For this purpose, 1 mL samples were fixed with glutaraldehyde (0.1%) and paraformaldehyde (1%) and preserved at −80 °C until analysis. For total bacterial counting by IFC, samples were thawed and stained with SYBR Green I nucleic acid gel stain (hereafter, SYBR Green) (0.01%) (490–498 nm, S-9430; Sigma-Aldrich, Saint Louis, MO, USA; ×10 dilution in DMSO of commercial stock), in dark for 10 min [54] before analysis by ISX. This fluorochrome allowed separation of bacterial cells from abiotic particles (e.g., detritus) (Figure 7B). 

In order to analyze the proportion of quiescent (dead) bacteria, an aliquot of 300 μL was collected and stained with SYTOX Green dead cell stain (0.001%) (488–530 nm, S-34860; Thermo Fisher Scientific, Waltham, MA, USA) immediately after sampling, rather than fixing. This stain does not penetrate live cells but only those with compromised plasma membranes. For this reason, it is used to determine cell viability [57] and to estimate membrane integrity in bacteria [58,59]. The optimal concentration (10^−3^ µM of the commercial solution) and time of incubation (10 min) of SYTOX Green were evaluated experimentally to adapt the protocol of Lebaron et al. [58]. Quantification of alive bacteria was obtained by the difference between total bacteria (SYBR Green stained bacteria) and dead bacteria (SYTOX Green stained bacteria).

### 4.4. Microalgal and Bacterial Growth Rates

Microalgae and bacteria exponential growth rates were calculated as μ (day^−1^) according to:$$µ=\ln\;\frac{N_{1}}{N_{0}}\;t$$
where N_0_ and N_1_ represent cell density at the start and the end of the exponential growth period, and t is the time between measurements (in days) [14].

### 4.5. PUA Sampling, Extraction, and Quantification

For analysis of the pPUA fraction, 50 mL samples were collected from the different cultures and the algal biomass was concentrated by centrifugation (5 min at 1110 rcf; 3000 rpm Mixtasel-BLT, SELECTA). Then, 2 mL of derivatization reagent (25 mM solution of *O*-(2,3,4,5,6-pentafluorobenzylhydroxylamine)hydrochloride; PFBHA, Fluka, Basel, Switzerland) in 100 mM Tris-HCl, pH 7, Trizma, Sigma, Steinheim, Germany) and 500 µL of 40 nM benzaldehyde (internal standard) were added to the resulting pellet. For mechanical disruption of the cells, the samples were sonicated by ultrasound (Bandelin Sonoplus, HD2070, 97%). The extraction was performed according to the protocol described in [20]. 

For analysis of the dissolved fraction (dPUA), 100 mL of the different cultures were sampled and filtered at low pressure through 2 µm Whatman GF/F filters to remove algal biomass. Then 1 mL of PFBHA reagent in Tris-HCl and 500 µL of the internal standard were added. After one hour at room temperature for complete derivatization, each sample was eluted through a LiChrolut^®^RP C-18 cartridge, previously washed with derivatization solution, by pumping an Eyela pump. The derivatized PUA was eluted from the cartridge with 4 mL of 10 mM PFBHA in methanol, collected in a glass vial, and incubated for at least one hour at ambient temperature to ensure complete derivatization of the aldehydes. The vials were then stored at −80 °C until extraction. For extraction, samples were transferred into a 100 mL glass separating funnel using an 8:4:8 ratio (hexane:methanol:water) following the protocol of [60].

The obtained extracts for both, pPUA and dPUA, were analyzed by GC-MS using an Agilent 7890A GC gas chromatography instrument (Agilent Technologies Inc., Santa Clara, CA, USA) coupled to either a Synapt G2 Q-TOF high-resolution mass spectrometer (Milford, MA, USA) with an atmospheric pressure ionization source (atmospheric pressure gas chromatography, APGC) in positive mode, or to a triple quadrupole spectrometer with an electron impact ionization (EI) source in multiple reaction monitoring (MRM) mode. Chromatographic separation of PUA was carried out using an HP-5MS column (30 m × 0.25 mm (i.d.) × 0.25 mm, 5% phenyl and 95% polydimethylsiloxane), at flow rate 1 mL min^−1^ and injection temperature of 280 °C. The temperature ramp used was: 70 °C for 1 min, incrementing at 35 °C min^−1^ up to 180 °C, and 4.50 °C min^−1^ up to 290 °C, and maintaining 290 °C for 8 min. TOF-MS analyses in API mode were performed in a range *m*/*z* = 50–1200, with a corona voltage of 2 kV, chamber temperature of 130 °C and a corona voltage between 10–40 V. PUA were identified comparing the retention times and exact molecular mass measurement (error less than 5 ppm) with those obtained from commercially available standard samples, 2*E*,4*E*-heptadienal (90%, Sigma-Aldrich Chemie GmbH, Steinheim, Germany), 2*E*,4*E*-octadienal (>96%, Sigma-Aldrich Chemie GmbH), and 2*E*,4*E*-decadienal (85%, Sigma-Aldrich Chemie GmbH, St. Louis, MO, USA). For the correct quantification of PUA, calibration lines were performed (1–7000 nM in hexane-2 mL) by comparing the intensity of the signals of the molecular ions of the standard samples. Different synthetic standard solutions of commercial HD, OD, and DD (from 1–15 nM, 15–400 nM, and 400–7000 nM) were used to obtain calibration curves to cover the wide range of molarities in the analysis of pPUA and dPUA. The calibration curves of pPUA and dPUA were constructed separately.

The results were obtained by plotting the peak area of each aldehyde against the area of the internal standard (benzaldehyde). Reproducibility and repeatability of this methodology were evaluated by reanalysis of the standard samples two weeks after the first analysis. The chromatograms were processed with MassLynx software (version 4.1, Waters, Milford, MA, USA).

### 4.6. Analysis of Fatty Acids

To characterize PUFA in *C. cryptica* at the two nutrient conditions assayed, a volume of 200 mL of the stock cultures at F2 and low P conditions were collected, both at exponential and stationary growth phase, following the procedure of [61]. After centrifugation (3500 rpm for 5 min), the pellets were extracted 5 times with a solution of acetone:methanol (1:1) and sonicated (ultrasound bath, 200 W–50Hz for 5 min). The combined extracts were filtered, evaporated under reduced pressure, and frozen until the analysis of fatty acid methyl esters (FAME). The transmethylation of fatty acids was carried out by treating the extracts with 1 mL of MeOH/HCl (10:1) and heating under reflux for 1 h. After cooling, the reaction was extracted with hexane (3 × 3 mL), and the organic layers were combined, dried over anhydrous MgSO_4_, and taken to dryness by rotary evaporation. Fatty acid methyl esters were analyzed by GC-MS on an Agilent Technologies 7890A GC gas chromatography instrument (Agilent Technologies Inc., Santa Clara, CA, USA) coupled to a triple quadrupole spectrometer with an electron impact ionization (EI) source at 70 eV and scanning the mass range *m*/*z* 50–550. Chromatographic separation of FAME was carried out using an HP-5MS column (30 m × 0.25 mm (i.d.) × 0.25 mm, 5% phenyl and 95% polydimethylsiloxane), at flow rate 1 mL min^-1^ and injection temperature of 280 °C. Fatty acid identification was established by comparing their retention time and mass spectrum with MS spectra of the commercial FAME standards Supelco 37 Component FAME Mix (ref. 47885-U, Sigma-Aldrich, Darmstadt, Germany) analyzed by GC-MS under the same conditions of FAME samples and using C17:0 as internal standard.

### 4.7. Statistical Analysis

Statistical significance was evaluated through one-way ANOVA test at a level of *p* < 0.05. Two-way ANOVAs were used to test for differences in pPUA and dPUA between phosphate availability, the presence or absence of bacteria, and the interaction of both factors. When appropriate, a Tukey HSD post hoc test was applied to examine all relevant pairwise comparisons between *C. cryptica* cultures. Spearman correlation analysis was conducted to examine the association between PUA concentrations (pPUA, dPUA, and types) and bacterial assemblage cell densities. All statistical analyses were performed with R-version 3.4.1.

## Figures and Tables

**Figure 1 marinedrugs-21-00571-f001:**
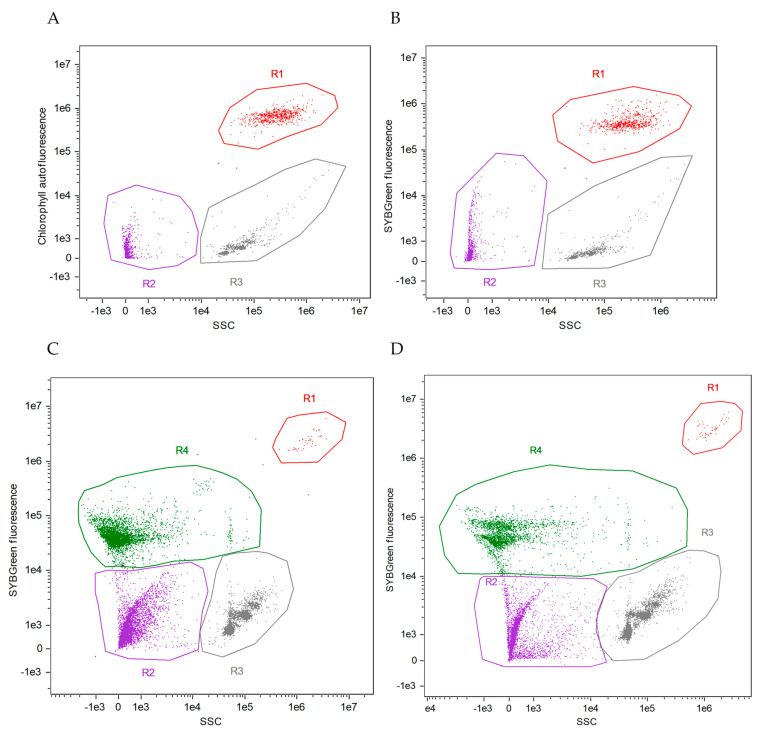
Type of cytograms of experimental cultures of *C. cryptica*. (**A**): axenic culture of *C. cryptica* without staining; (**B**): axenic culture of *C. cryptica* stained with SYBR Green, confirming that no bacteria were present in the culture; (**C**): non-axenic culture of *C. cryptica* with autochthonous bacteria (N-culture) stained with SYBR Green; (**D**): non-axenic culture of *C. cryptica* with introduced bacteria (I-culture) stained with SYBR Green. Sample key: R1 = *C. cryptica*; R2 = detritus and cellular residues; R3 = aggregated beads; R4 = heterotrophic bacteria.

**Figure 2 marinedrugs-21-00571-f002:**
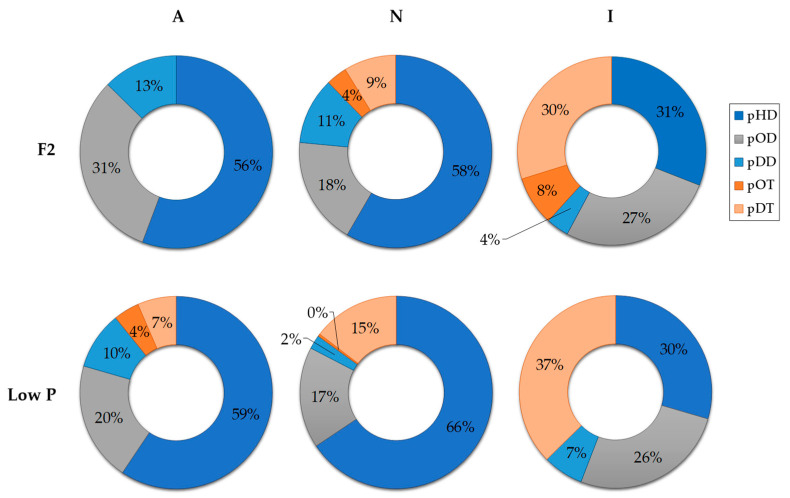
Sector diagram of the averaged (*n* = 5) relative percentage of pPUA types detected in *C. cryptica* cultures quantified at late exponential growth phase under F2 and low P conditions. Sample key: A-cultures = axenic cultures of *C. cryptica*; N-cultures = *C. cryptica* co-cultured with autochthonous heterotrophic bacteria; I-cultures = *C. cryptica* co-cultured with introduced heterotrophic bacteria. pHD = particulate *2E,4E/Z*-heptadienal; pOD = particulate *2E,4E/Z*-octadienal; pDD = particulate *2E,4E/Z*-decadienal; pOT = particulate *2E,4E/Z,7*-octatrienal; pDT = particulate *2E,4E/Z,7Z*-decatrienal. pPUA data are normalized by cell density (*n* = 5).

**Figure 3 marinedrugs-21-00571-f003:**
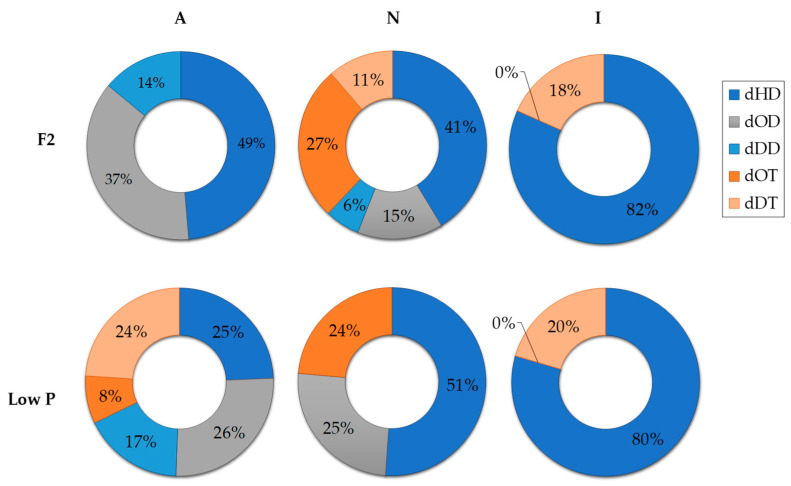
Sector diagram of the relative percentage of different dPUA types detected in *C. cryptica* cultures quantified in late exponential growth phase at the two phosphate availability conditions assayed (F2 and low P) (*n* = 5). Sample key: A-cultures = axenic cultures of *C. cryptica*; N-cultures = *C. cryptica* co-cultured with autochthonous heterotrophic bacteria; I-cultures = *C. cryptica* co-cultured with introduced heterotrophic bacteria.; dHD = dissolved *2E,4E/Z*-heptadienal; dOD = dissolved *2E,4E/Z*-octadienal; dDD = dissolved *2E,4E/Z*-decadienal; dOT = dissolved *2E,4E/Z,7*-octatrienal; dDT = dissolved *2E,4E/Z,7Z*-decatrienal.

**Figure 4 marinedrugs-21-00571-f004:**
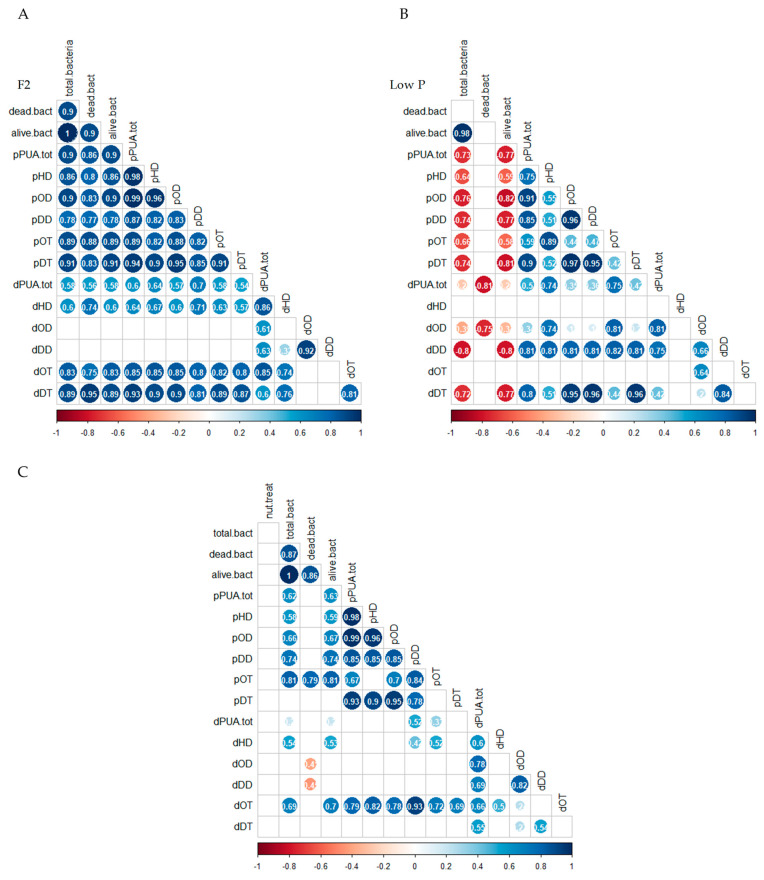
Correlograms illustrating Spearman’s rank correlation coefficients between pPUA, dPUA, and bacterial abundance at F2 conditions (**A**), pPUA, dPUA, and bacterial abundance at low P conditions (**B**) and pPUA, dPUA, bacterial abundance and phosphate treatment (**C**). The circles show significant pairwise correlations (*p*-value < 0.05), and the color intensity and circle sizes represent r coefficient values, red marks are negative correlation, and blue marks are positive correlation. Sample key: p: particulate; d: dissolved; HD = *2E,4E/Z*-heptadienal; OD = *2E,4E/Z*-octadienal; DD = *2E,4E/Z*-decadienal; OT = *2E,4E/Z,7*-octatrienal; DT = *2E,4E/Z,7Z*-decatrienal.

**Figure 5 marinedrugs-21-00571-f005:**
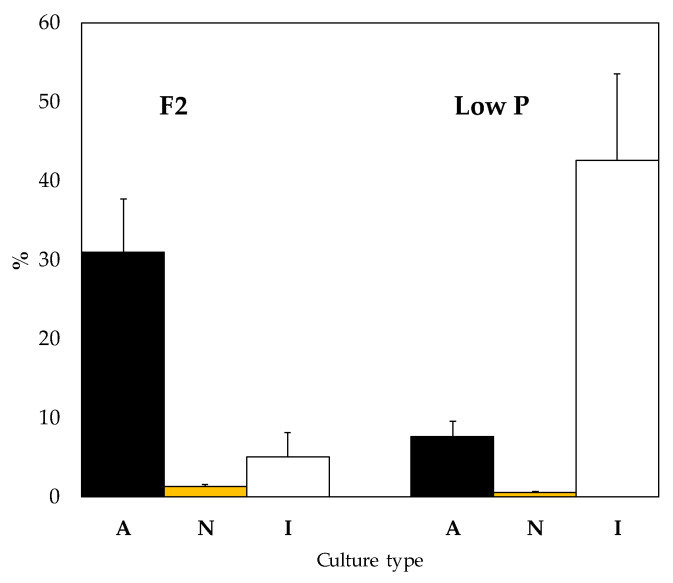
Percentage of dPUA released in the *C. cryptica* cultures quantified in late exponential growth phase at the two phosphate availability conditions assayed (F2 and low P). (*n* = 5). Sample key: A-cultures = axenic cultures of *C. cryptica*; N-cultures = *C. cryptica* co-cultured with autochthonous heterotrophic bacteria; I-cultures = *C. cryptica* co-cultured with introduced heterotrophic bacteria.

**Figure 6 marinedrugs-21-00571-f006:**
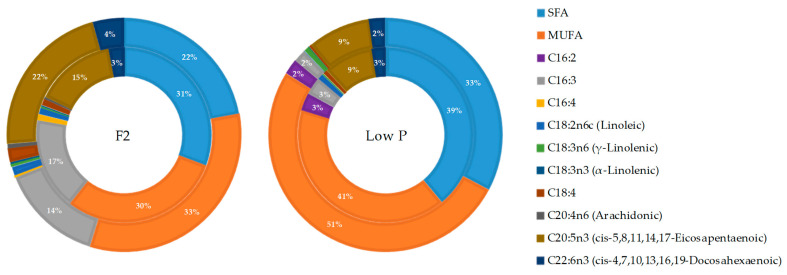
Sector diagram of the different FAME detected in *C. cryptica* stock cultures, at the two phosphate availability conditions assayed (F2 and low P). The inner pie chart shows the results obtained at exponential growth phase results, and the outer one shows the results obtained at late exponential growth phase. (*n* = 3). Percentages less than 2% are not shown.

**Figure 7 marinedrugs-21-00571-f007:**
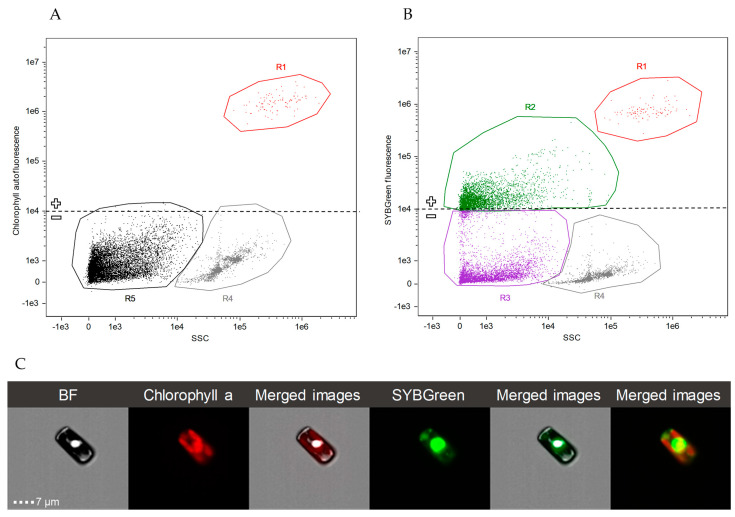
Type of cytogram of an experimental *C. cryptica* N-culture. (**A**): Representative dot plot of chlorophyll autofluorescence vs. SSC, both in arbitrary units, that permits distinction of photosynthetic cells (+chlorophyll autofluorescence). Diatom population is clearly distinguished from detritus and aggregated calibration beads of the ISX by positive chlorophyll autofluorescence (arbitrary units ≥ 10^4^) and SSC. (**B**): A representative dot plot of SYBR Green fluorescence vs. SSC, both in arbitrary units, that permits distinction between biotic (+DNA stained) and abiotic particles (−DNA stained). Distinct subpopulations are visible on the plot, where R1 = *C. cryptica* cells; R2 = heterotrophic bacteria with SYBR Green staining; R3 = detritus and cellular residues; R4 = aggregated beads; R5 = heterotrophic bacteria, detritus and cellular residues without SYBR Green staining; (**C**): IFC microphotographs of a *C. cryptica* from the bright field (BF), chlorophyll red autofluorescence that defines chloroplast morphology (chlorophyll a) and green fluorescence that defines the nucleus morphology (SYBR Green fluorescence) and merged images.

**Figure 8 marinedrugs-21-00571-f008:**
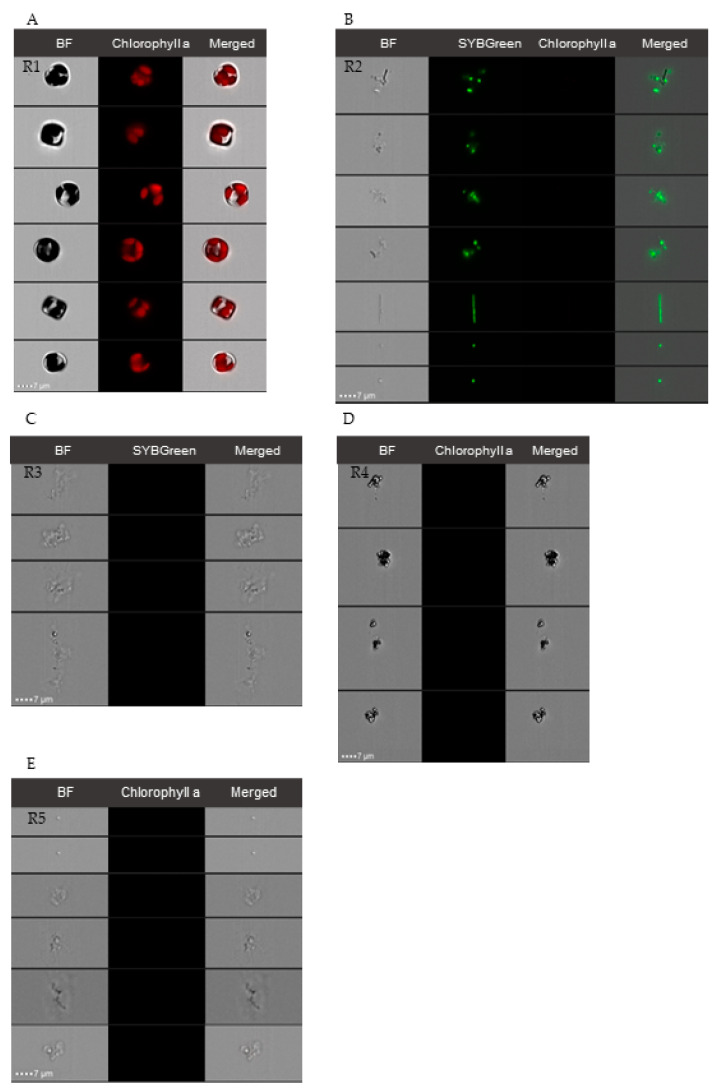
IFC representative microphotographs at 60× associated with dots selected in experimental N-cultures of *C. cryptica*; (**A**): different alive cells of *C. cryptica* with chlorophyll red autofluorescence that defines chloroplasts morphology (R1 in Figure 7); (**B**): different heterotrophic bacteria stained with SYBR Green, aggregated and individual cells are shown (R2 in Figure 7); (**C**): detritus and cellular residues, without green signal (R3 in Figure 7); (**D**): aggregated beads, without red signal (R4 in Figure 7); (**E**): bacteria, detritus, and cellular residues, without red signal (R5 in Figure 7).

**Table 1 marinedrugs-21-00571-t001:** Growth rates (μ) of *C. cryptica* cultures (day^−1^) at the two phosphate availability conditions assayed (F2 and low P). Data are expressed as mean ± standard deviation (SD) of *n* = 5. Sample key: A-cultures = axenic cultures of *C. cryptica*; N-cultures = *C. cryptica* co-cultured with autochthonous heterotrophic bacteria; I-cultures = *C. cryptica* co-cultured with introduced heterotrophic bacteria.

	*C. cryptica* Cultures			Heterotrophic Bacteria	
Phosphate Availability Treatment	A	N	I	Autochthonous Bacteria	Introduced Bacteria
F2	0.35 ± 0.06	0.52 ± 0.08 ***	0.82 ± 0.21 **	0.22 ± 0.11	2.13 ± 0.10 ***
Low P	0.42 ± 0.02	0.53 ± 0.08 *	0.73 ± 0.04 ***	0.13 ± 0.04	1.57 ± 0.15 ***

Significant differences (one-way ANOVA) between axenic cultures and non-axenic cultures of *C. cryptica* and between autochthonous and introduced bacteria are indicated with asterisks (* *p*-value < 0.05, ** *p*-value < 0.01, and *** *p*-value < 0.001).

**Table 2 marinedrugs-21-00571-t002:** Total pPUA (particulate PUA) concentrations (mean ± standard deviation) of *C. cryptica* cultures quantified in late exponential growth phase at the two phosphate availability conditions assayed (F2 and low P). pPUA is expressed as fmol cell^−1^ (*n* = 5). Sample key: A-cultures = axenic cultures of *C. cryptica*; N-cultures = *C. cryptica* co-cultured with autochthonous heterotrophic bacteria; I-cultures = *C. cryptica* co-cultured with introduced heterotrophic bacteria.

Phosphate Availability	*C. cryptica* Cultures	Total pPUA (fmol cell^−1^)
F2	A	0.02 ± 0.00
	N	2.42 ± 0.49 ***
	I	0.16 ± 0.07 **
Low P	A	0.24 ± 0.05
	N	3.46 ± 0.43 ***
	I	0.02 ± 0.01 ***

Statistically significant differences (one-way ANOVA) between axenic cultures and non-axenic cultures of *C. cryptica* are indicated with asterisks (** *p*-value < 0.01, and *** *p*-value < 0.001). Detailed statistical data are included in Appendix A.

**Table 3 marinedrugs-21-00571-t003:** pPUA-type concentrations (mean ± standard deviation) of *C. cryptica* cultures quantified in late exponential growth phase at the two phosphate availability conditions assayed (F2 and low P). pPUA is expressed as fmol cell^−1^ (*n* = 5). Sample key: A-cultures = Axenic cultures of *C. cryptica*; N-cultures = *C. cryptica* co-cultured with autochthonous heterotrophic bacteria; I-cultures = *C. cryptica* co-cultured with introduced heterotrophic bacteria. pHD = particulate *2E,4E/Z*-heptadienal; pOD = particulate *2E,4E/Z*-octadienal; pDD = particulate *2E,4E/Z*-decadienal; pOT = particulate *2E,4E/Z,7*-octatrienal; pDT = particulate *2E,4E/Z,7Z*-decatrienal. n.d. = non-detected.

Phosphate Availability	*C. cryptica* Cultures	pHD	pOD	pDD	pOT	pDT
F2	A	0.01 ± 0.00	0.005 ± 0.00	0.002 ± 0.00	n.d.	n.d.
	N	1.41 ± 0.37 ***	0.44 ± 0.09 ***	0.27 ± 0.05 ***	0.08 ± 0.05	0.21 ± 0.04
	I	0.05 ± 0.02 *	0.04 ± 0.02 **	0.006 ± 0.00 **	0.01 ± 0.01	0.05 ± 0.03
Low P	A	0.14 ± 0.01	0.05 ± 0.01	0.02 ± 0.01	0.01 ± 0.01	0.015 ± 0.00
	N	2.27 ± 0.21 ***	0.59 ± 0.09 ***	0.08 ± 0.03 *	0.01 ± 0.00	0.51 ± 0.29 **
	I	0.01 ± 0.00 ***	0.01 ± 0.00 ***	0.002 ± 0.00 ***	n.d.	0.01 ± 0.01 *

Statistically significant differences (one-way ANOVA) between axenic cultures and non-axenic cultures of *C. cryptica* are indicated by asterisk marks. Level of significance: * *p*-value < 0.05, ** *p*-value < 0.01, and *** *p*-value < 0.001. Detailed statistical data are included in Appendix A.

**Table 4 marinedrugs-21-00571-t004:** Total dPUA concentrations (mean ± standard deviation) of *C. cryptica* cultures quantified in late exponential growth phase at the two phosphate availability conditions assayed (F2 and low P) (*n* = 5). Sample key: A-cultures = Axenic cultures of *C. cryptica*; N-cultures = *C. cryptica* co-cultured with autochthonous heterotrophic bacteria; I-cultures = *C. cryptica* co-cultured with introduced heterotrophic bacteria.

Phosphate Availability	*C. cryptica* Cultures	Total dPUA (nM)
F2	A	4.79 ± 1.08
	N	9.72 ± 2.13 **
	I	2.03 ± 1.65 *
Low P	A	8.39 ± 2.19
	N	4.81 ± 0.32 **
	I	3.64 ± 1.12 **

Significant differences (one-way ANOVA) between axenic cultures and non-axenic cultures of *C. cryptica* are indicated with asterisks (* *p*-value < 0.05, ** *p*-value < 0.01). Detailed statistical data are included in Appendix A.

**Table 5 marinedrugs-21-00571-t005:** dPUA-type concentrations (mean ± standard deviation) of *C. cryptica* cultures quantified in late exponential growth phase at the two phosphate availability conditions assayed (F2 and low P) (*n* = 5). Sample key: A-cultures = axenic cultures of *C. cryptica*; N-cultures = *C. cryptica* co-cultured with autochthonous heterotrophic bacteria; I-cultures = *C. cryptica* co-cultured with introduced heterotrophic bacteria.; dHD = dissolved *2E,4E/Z*-heptadienal; dOD = dissolved *2E,4E/Z*-octadienal; dDD = dissolved *2E,4E/Z*-decadienal; dOT = dissolved *2E,4E/Z,7Z*-octatrienal; dDT = dissolved *2E,4E/Z,7Z*-decatrienal. n.d. = not detected.

Phosphate Availability	*C. cryptica* Cultures	dHD	dOD	dDD	dOT	dDT
F2	A	2.33 ± 0.70	1.79 ± 0.25	0.67 ± 0.21	n.d.	n.d.
	N	4.01 ± 1.14 *	1.43 ± 0.08 *	0.58 ± 0.12	2.59 ± 0.38	1.10 ± 0.48
	I	1.65 ± 1.40	n.d.	n.d.	n.d.	0.37 ± 0.26
Low P	A	2.04 ± 0.54	2.20 ± 0.73	1.45 ± 0.35	0.69 ± 0.13	2.01 ± 0.57
	N	2.46 ± 0.32	1.22 ± 0.01 *	n.d.	1.14 ± 0.00 ***	n.d.
	I	2.89 ± 1.05	n.d.	n.d.	n.d.	0.75 ± 0.16 **

Significant differences (one-way ANOVA) between axenic cultures and non-axenic cultures of *C. cryptica* were found. An asterisk marks result of significant value where * *p*-value < 0.05, ** *p*-value < 0.01, and *** *p*-value < 0.001. Detailed statistical data are included in Appendix A.

**Table 6 marinedrugs-21-00571-t006:** Percentages of FAME (fatty acid methyl ester) in *C. cryptica* stock cultures quantified in exponential growth phase and late exponential growth phase at two phosphate availability conditions assayed (F2 and low P).

Phosphate Availability	FAME	% TFA (Exponential)	% TFA (Late Exponential)
F2	SFA	30.82	22.16
	MUFA	29.85	32.62
	PUFA	39.32	45.22
	EPA	14.90	21.92
	DHA	2.93	4.31
Low P	SFA	38.93	32.75
	MUFA	40.65	51.09
	PUFA	20.42	16.16
	EPA	9.18	8.75
	DHA	2.60	2.24

## Data Availability

Data are contained within this article and Appendix A.

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
