# Peer review of "Polyunsaturated Aldehydes Profile in the Diatom Cyclotella cryptica Is Sensitive to Changes in Its Phycosphere Bacterial Assemblages"

_marinedrugs, 2023, doi:10.3390/md21110571_

Round 1

Reviewer 1 Report

Comments and Suggestions for Authors

A highly interesting study but since the authors did not use metabarcoding to identify the bacteria its merit is decreased. I personally think that statements such as "PUA as a specific organic matter... to attract beneficial bacteria for constructing its own phycosphere, for more beneficial growth" is not directly confirmed by the data obtained and thus the text has to be formulated far more cautiously. 

Comments on the Quality of English Language

The quality of the English writing is good except in some cases more work has to be done.

Reviewer 2 Report

Comments and Suggestions for Authors

General Comments:

In this study, Hernanz-Torrijos et al investigate how polyunsaturated aldehydes (PUA) profile is affected by changes in the bacterial communities within the phycosphere of the diatom Cyclotella cryptica. This is an interesting and original piece of work.

There are only a handful of minor issues to resolve prior to publication. The authors should reconsider the structure of their manuscript according to the standards of the journal. Please adjust references to the Marine Drugs journal format. Abbreviations: the first time, the full word should be used with the abbreviation between brackets; afterwards, only the abbreviation should be used. Abbreviations should be unified throughout the whole document. The placement of the figures could be optimized for better clarity.

Specific Comments

1.  Lines 58-60 : “….diatom–bacteria interactions in the phycosphere has been less studied…”,

There have been some recent advancements in the study of PUA-mediated diatom-bacterial interactions. In fact, two recent publications specifically address the potential role of PUAs in mediating these interactions within the phycosphere. I have attached the relevant references for your consideration.

Ref1: Science of the Total Environment, 2023, doi: 10.1016/j.scitotenv.2023.166518; 

Ref2: Biogeosciences, 2021, doi: 10.5194 /bg-18-1049-2021.

2.  Lines 61-64 : “…PUAs…”, I would suggest standardizing the abbreviation to "PUA" throughout the manuscript, including on lines 61 and 64.

3.  Line 97 : “…with F2 conditions…”, Could you please clarify if you are referring to "F2 culture conditions" or "F2 nutrition conditions"?

4.  Lines 116, 128, 186, 264, 299, 313,596 : “…pPUA…dPUA….DF…FAME…MRM…”, Abbreviations: the first time, the full word should be used with the abbreviation between brackets.

5.  Lines 146-161, 217-229, Concerning the component types of pPUA and dPUA, it's necessary to provide the full names for each component of particulate PUA and dissolved PUA. For instance, "particulate 2E,4E/Z‐heptadienal (pHD)" and "dissolved 2E,4E/Z‐heptadienal (dHD)". Please continue in this manner for other components as well.

6.  Lines 468, 532, 557, 562, Could you clarify the distinction between your descriptions of staining with SYBR GREEN and staining with SYTOX GREEN?

7.  Lines 368-369, 393 : “…. These PUAs have been well documented as the dominant bioactive PUAs released by diatoms in the past…”,

Recent advancements have been made in the study of in-situ diatom PUA . I have attached the relevant references for your consideration.

Ref1: Science of the Total Environment, 2023, doi: 10.1016/j.scitotenv.2023.166518; 

Ref2: Biogeosciences, 2021, doi: 10.5194 /bg-18-1049-2021.

Ref3: Journal of Geophysical Research: Biogeosciences, 2021, doi: 10.1029/2020JG005808; 

Ref4: Progress in Oceanography, 2016, doi: 10.1016/j.pocean.2016. 07.010.

8.  Line 471 : For the experimental design section, I would suggest including a flowchart outlining the experimental process? This would enable readers to more quickly grasp the specific steps and sampling methods the authors utilized. A few helpful references for this format could be found below.

Ref1: Biogeosciences, 2021, doi: 10.5194 /bg-18-1049-2021.

Ref2: Marine Biology, 2017, 164, 1-11.

9.  Lines 537-535 : According to the formatting guidelines of this journal, it appears that the "Materials and Methods" section is typically positioned at the end. Figures 7 and 8 might be better placed in the supplementary materials, allowing for a clearer emphasis on the earlier figures and tables in the main text.

10.  Line 580 : ”… 100 mL of the different cultures…”, In the 250 mL culture system, how many times did you measure the dissolved PUA?

11.  Line 582 : ”… Tris and …”, This sentence may be revised as: Tris-HCl and …

12.  Lines 607 : The calibration curve might distinguish between two PUA types: pPUA and dPUA. Could you briefly introduce the method of the calibration curve?

13.  Lines 669,762,769773: Please adjust references to the Marine Drugs journal format.

Reviewer 3 Report

Comments and Suggestions for Authors

The present paper is focused on the variation of PUAs profile of the centric diatom Cyclotella cryptica under different conditions, which regards the presence/absence of bacteria and the typology of the bacterial consortium.

Overall, the work is well organised and the experimental set-up is clear. In the discussion, the authors clearly described their hypotheses, and highlighted the consistency of their results with previous works. Moreover, they highlighted the “limit” of their work, e.g. the impossibility to conclude if PUAs amount depend on signals from bacteria or if diatoms consume bacterial metabolites.

I have some observations the authors should consider to improve the manuscript:

It seems that the paper was initially written by inserting the M&M section before the results, and that the order of the paragraphs has been changed at a later time. Since the Results section precedes the M&M one, the authors should explain, before presenting their results, the reason why they performed the analysis of PUAs on P. tricornutum. It becomes clear only in the M&M section.

The authors should specify also in the text (before the insertion of the table 4) the meaning of A-, N- and I- cultures.

Moreover, I will appreciate the insertion of details regarding axenic cultures: specifically, the authors should describe how did they obtain axenic strains (antibiotic treatments in lab? Cultures purchased without bacteria?).

I also found some typing errors/imperfections listed below:

Line 17: replace that with which

line 26: replace signature with sign

line 38/39: replace in particularly with in particular

line 355: replace interaction with interactions

line 379: replace two point with point

line 573: please replace, if possible, rpm with rcf values, or indicate both values

Comments on the Quality of English Language

Minor editing of English language required

Reviewer 4 Report

Comments and Suggestions for Authors

General comments:

 This work analyses the importance of PUA as infochemicals for marine bacteria. It can provide some new information on these bioactive chemicals role in the ecosystems.

It is well written, and the hypothesis seem to be clearly stated and the results clearly interpreted. Therefore, my suggestions are mainly minor revisions. The major revision request is just because of some doubts about methodology details.

Introduction:

Comment 1 : line51 – bottom-up? or is it top-down? there are several studies in which it is seen that predation of diatoms by zooplankton leads to the production of PUA which in turn is toxic to the zooplankton, as is the case of copepods. In these cases, as the PUA is produced in response to the grazing wouldn't it be a top-down response?

Comment 2: line 62 and 64 – there are references out of format in these lines. At first I thought it was a matter of the references were inside the main text but further ahead the correct format was kept in these same conditions ex: “as has been documented by [25]” in line 421.

Results:

Comment 3: The tables 3 and 6 are the only that have the ANOVA test values. Why? I understand that there are already many data tables and the authors want to avoid more tables but then please add them to the supplementary material and keep consistency (table 3 and 6 should also go), or in turn put them all in the main document.

Comment 4: Table 3 and 6 – Please specify in the table caption that the post-hoc test was Tukey test (according to the methodology). Also specify that the Tukey test was made for the interactions (for what I understand of your table). I understand that this may be interpreted from the table statistics, but this alteration will make the table easier for readers to understand. (Even if the authors choose to send it to the supplementary materials).

Methodology:

Comment 5: line 479 – what are the justifications for the phosphate value of the low P treatment? Also, for both scenarios, please add the N and Si concentrations for comparison with the P value.

Comment 6: lines 484-494 - Do the authors have any idea of the bacteria taxonomy (gender or families) that were in each culture? How different were the bacteria composition of the cultures?

Comment 7: line 491-492 – I believe this information is mistaken. GF/F is a glass fiber filter (not polycarbonate) and has a porosity of 0.7 µm (and thus retain both bacteria – which length is usually higher than 1 µm – and algae). Please correct or provide more information about this methodology.

Comment 8: line 637: “…in some variables…” – please specify the variables analysed with the ANOVA.

Round 2

Reviewer 1 Report

Comments and Suggestions for Authors

The authors did a very fine job - Thank you very much for your clarifications. No further comments from my side.

Reviewer 4 Report

Comments and Suggestions for Authors

The authors have satisfactorily respond to all my queries and corrected the issues in the methodology. Therefore I believe the paper is ready for acceptance.